# CONTINUITY-DRIVEN POSE ESTIMATION FOR VIDEOS

## ABSTRACT

Video-based pose estimation plays a critical role in understanding human actions and enabling effective human-computer interaction. By exploiting temporal information from video frames, it enhances the localization of human keypoints. Previous feature-fusion methods often rely on a frozen single-frame backbone trained on individual frames, followed by a network to learn temporal information from video sequences. Consequently, these approaches fail to capture the temporal continuity between frames at the backbone network level, thereby restricting the network's capacity to effectively learn and leverage sequential information. In this paper, we introduce a novel approach to supervise continuity in the whole video pose estimation model from two perspectives: semantic continuity and pixel-wise keypoint distribution continuity. To this end, we propose a Semantic Alignment Space, where a semantic alignment encodes feature maps from different frames into this space, ensuring continuous supervision of the encoded representations. To further maintain pixel-wise keypoint distribution continuity, we introduce the Trajectory Probability Difference Integration method, which minimizes the trajectory difference expectation across frames. Additionally, to better capture temporal dependencies, we present a Multi-frame Heatmap Fusion structure that aggregates heatmaps from adjacent frames for a more refined output. Extensive experiments on the PoseTrack17, PoseTrack18, and PoseTrack21 datasets demonstrate the effectiveness of our approach, consistently achieving state-of-the-art results.

## 1 INTRODUCTION

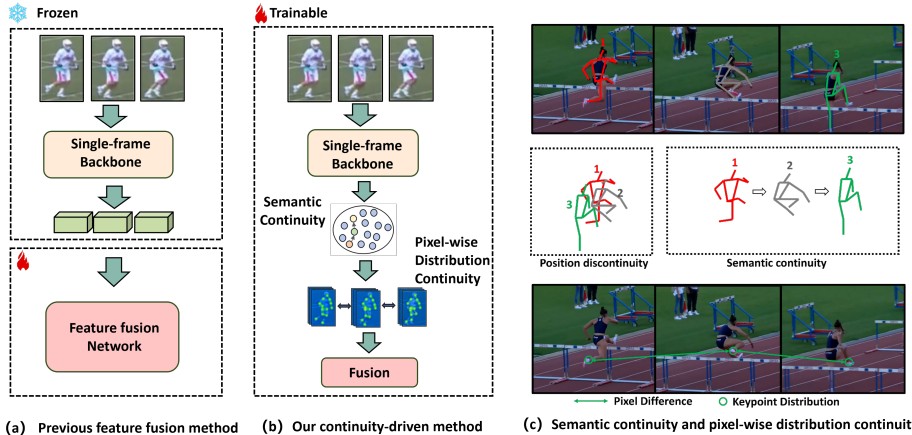

Figure 1: Overview of the continuity-driven pose estimation method.

Human pose estimation, a fundamental task in computer vision, aims to accurately predict the co-ordinates of keypoints corresponding to different parts of an individual's body. This technique has garnered significant attention in recent years due to its diverse applications, including action recognition, human-computer interaction, and motion analysis.

The task of video pose estimation focuses on predicting the human keypoint coordinates within a sequence of video frames. Capturing temporal information embedded in the video frames is therefore

crucial for precise pose estimation over time. To address this, existing approaches utilize Recurrent Neural Networks (RNNs) or 3D convolutional networks to process video sequences Luo et al. (2018); Wang et al. (2020). However, these methods are often computationally expensive and yield suboptimal performance. Alternatively, techniques such as optical flow Pfister et al. (2015); Song et al. (2017) have been employed to compute additional flow data, but they tend to be inaccurate, especially in the presence of motion blur. More recent methods Feng et al. (2023a); Liu et al. (2022; 2021) lean towards multi-frame feature fusion. In this approach, a pre-trained backbone network extracts features from individual frames, and a dedicated fusion network subsequently integrates the temporal context. However, this method has a significant limitation: the backbone, trained on isolated frames, is unable to exploit the temporal consistency inherent in video sequences. Consequently, the feature fusion process is disconnected from the backbone, which hampers the model's ability to fully capture and utilize the temporal dynamics of human motion. Additionally, due to the lack of supervision regarding frame-to-frame continuity, these techniques often result in pose discontinuities, reducing the model's overall performance.

To overcome these limitations, as shown in figure 1, we propose a novel framework that enforces temporal continuity for the whole video-based pose estimation models from two complementary perspectives: semantic continuity and pixel-wise keypoint distribution continuity. These aspects are critical for ensuring smooth pose transitions and preventing abrupt, unrealistic changes in keypoint locations between frames. Unlike previous approaches that only supervise the feature fusion process after freezing the backbone, we supervise the entire video pose estimation network by leveraging the inherent temporal continuity in the video sequence. This allows the backbone to learn temporal information alongside the feature fusion network.

More specifically, semantic continuity refers to the smooth, consistent representation of human poses despite changes in viewpoint, body configuration, or camera motion. While keypoint locations may shift dramatically across frames, the underlying semantic meaning of human actions remains consistent. For example, in figure 1 (c), due to viewpoint transformations and human motion, the coordinates of actions 1, 2, and 3 in consecutive frames are not continuous. The position of action 2 occurs before that of action 1, while the position of action 3 occurs after action 1. However, semantically, actions 1, 2, and 3 form a continuous sequence. To capture this, we introduce the Semantic Alignment Space, which encodes feature representations from different frames into a shared, pose-invariant latent space. This space ensures that semantically similar frames remain close in representation, even if keypoint positions vary. We apply contrastive loss to pull together the representations of adjacent frames while pushing apart those of distant frames, thereby preserving semantic continuity throughout the sequence. To ensure position invariance in this space, we apply transformations such as rotation and scaling and supervise the model to produce consistent results before and after the transformation. Since the semantic space is pose-invariant, we also use another encoder to capture positional transformations explicitly. A decoder then combines the semantic and positional encodings to generate keypoint probability heatmaps for the video sequence.

Beyond semantic continuity, we also supervise the continuity of keypoint probability distributions by the pixel continuity of frames, because it is noticed that the pixel differences for the same keypoint between consecutive frames can be considered approximately invariant, given that lighting conditions are typically stable over short periods. To this end, we introduce the Trajectory Probability Difference Integration method. As shown in figure 1 (c), since the model generates probability distributions for keypoints, we can calculate the expected difference for each keypoint across frames using keypoint distributions. By minimizing the integral of the keypoint trajectory probability differences of frame sequence, we ensure smoother and more accurate keypoint predictions across video. Notably, while optical flow methods Pfister et al. (2015); Song et al. (2017) also calculate pixel differences between adjacent frames, they compute the optical flow field as input to the network to capture temporal information. In contrast, our approach incorporates pixel differences into the loss function to supervise the keypoint distribution in the video pose estimation network. Rather than computing the difference for every pixel, we focus on positions with higher keypoint probability distributions. Furthermore, our method is only applied during training and does not require extra computation during inference, making it more efficient than optical flow-based approaches.

To further enhance the temporal modeling, we propose a Multi-frame Heatmap Fusion module. This mechanism aggregates heatmaps from adjacent frames by alternately using spatial attention and temporal self-attention, creating a refined output that incorporates temporal context. By fusing

information across multiple frames, the model generates more stable and accurate keypoint predictions, improving overall pose estimation performance.

Extensive experiments on benchmark datasets, including PoseTrack17, PoseTrack18, and PoseTrack21, demonstrate the effectiveness of our approach. Our method consistently outperforms state-of-the-art models, showcasing its ability to better capture temporal dependencies and deliver coherent pose estimations across video frames.

In summary, our main contributions are outlined as follows:

- We identify two critical types of continuity for video pose estimation: semantic space continuity and pixel-wise keypoint distribution continuity. To address these, we design a novel approach that supervises both aspects within the pose estimation model. To ensure semantic continuity across frames, we introduce the Semantic Alignment Space. Furthermore, we propose the Trajectory Probability Difference Integration method to enforce smooth pixel-wise keypoint distribution continuity throughout the video sequence.

- We propose a Multi-frame Heatmap Fusion module, which merges pose heatmap sequences to generate a new fusion heatmap, enhancing the model's performance.

## 2 RELATED WORKS

### 2.1 IMAGE-BASED HUMAN POSE ESTIMATION.

Image-based human pose estimation seeks to precisely infer the coordinates of key points on individuals within images. With the evolution of artificial neural networks, various deep-learning approaches are employed for image-based human pose estimation. These methodologies can be broadly categorized into two paradigms: bottom-up methods and top-down approaches.

The bottom-up approach is proposed in Deepcut Pishchulin et al. (2016) and significantly improved in OpenPose Cao et al. (2017). These methods detect all human keypoints in an image at once and cluster them into persons. Most bottom-up methods Cao et al. (2017); Newell et al. (2017); Cheng et al. (2020); Jin et al. (2022); Cai (2021) are based on the heatmap, and use Part Affinity Fields for keypoint clustering. Alternatively, the top-down approaches Li et al. (2023); Xiao et al. (2018); Chen et al. (2018); Newell et al. (2016); Sun et al. (2019); He et al. (2017); Kamel et al. (2020) decompose multi-person pose estimation into two distinct stages. Initially, a human detector is utilized to detect each individual within the image. Following this, the patches within the bounding boxes produced by the human detector are cropped and sequentially input into the single-person pose estimation network. Although this method introduces an additional processing step, compared to the bottom-up approach, it typically exhibits a noticeable advantage in terms of performance.

### 2.2 VIDEO-BASED HUMAN POSE ESTIMATION.

Regarding video-based human pose estimation, early methods primarily relied on image-based approaches, which, unfortunately, fell short due to their inability to account for temporal dependencies between frames. Lately, optical flow-based strategies Song et al. (2017); Pfister et al. (2015) generate optical flow between successive frames, leveraging these optical flows as motion indicators to enhance predicted pose heatmaps. However, such flow generation is computationally expensive and demonstrates vulnerability under significant image quality deterioration. Luo et al. (2018) uses RNN to capture temporal and spatial information, directly predicting the keypoint heatmap sequences for videos. A noteworthy alternative approach is the utilization of 3D Convolutional Neural Networks (3DCNNs) Wang et al. (2020) or deformable convolution Liu et al. (2022) to integrate heatmaps across frames, leading to improved heatmap quality. Some techniques Liu et al. (2022); Feng et al. (2023a) also incorporate multi-frame feature fusion, thereby bolstering video pose estimation accuracy. Recent advanced method Feng et al. (2023b) integrates transformer-based designs with diffusion models, evidencing substantial enhancements in pose estimation results.

## 3 METHODOLOGY

In this section, we provide an overview of our proposed methodology. First, we describe our approach for enforcing continuity in video-based pose estimation through two complementary mechanisms: semantic alignment and pixel-wise keypoint distribution continuity. We introduce the Semantic Alignment Space to maintain semantic consistency across frames and the Trajectory Probability Difference Integration method to enforce temporal continuity in the keypoint distribution heatmaps.

After introducing the Trajectory Probability Difference Integration, we discuss the overall network architecture, including the Multi-frame Heatmap Fusion module, which effectively fuses information across frames to improve pose estimation.

Finally, we explain the training and loss function for our video pose estimation network.

### 3.1 PROBLEM SETTING

The challenge in video-based pose estimation lies in capturing temporal information embedded in sequential frames, which can be used to enhance pose estimation accuracy over time. In contrast to image-based pose estimation, which processes isolated frames, video-based methods utilize a sequence of frames to model temporal dynamics.

We denote a sequence of consecutive frames around a target frame $I_t$ as $x = \{I_{t-T}, \ldots, I_{t+T}\}$, where $T$ is the temporal window size. Our goal is to leverage these temporal dynamics to improve the keypoint predictions at keyframe $I_t$. Following a Top-Down approach, we first apply a human detector to each frame to extract bounding boxes for each detected person. These bounding boxes are used to form a personalized input sequence $x^p = \{I_{t-T}^p, \ldots, I_{t+T}^p\}$ for each individual. This personalized sequence is then passed into our model to predict the keypoint coordinates in $I_t^p$.

### 3.2 SEMANTIC ALIGNMENT SPACE

To model the semantic continuity of human body poses across video frames, we introduce the Semantic Alignment Space, a position-invariant latent space that preserves pose semantics across frames. The core idea is that while keypoint positions may vary due to motion or changes in viewpoint, the overall human action should remain semantically consistent. By aligning frame features in this space, we ensure that poses in consecutive frames are semantically coherent.

To encode features into this space, we use a Semantic Alignment Encoder $E_Z$, which consists of Multi-Head Self-Attention (MHSA) and MLP-Mixer blocks. Given a feature map $F \in \mathbb{R}^{h \times w \times c}$, where $h$, $w$, and $c$ denote the height, width, and number of channels, respectively, the encoder outputs an $M$-dimensional semantic embedding $Z$:

$$Z = E_Z(F). \tag{1}$$

We employ a contrastive loss to enforce semantic continuity in this space. The loss function ensures that the embeddings of temporally close frames are more similar than those of distant frames:

$$\mathcal{L}_c = \max(0, \|Z_{t-\delta} - Z_{t+\delta}\|_2 - \|Z_{t-\delta} - Z_t\|_2 + \alpha) + \max(0, \|Z_{t-\delta} - Z_{t+\delta}\|_2 - \|Z_t - Z_{t+\delta}\|_2 + \alpha), \tag{2}$$

where $\delta$ is the frame interval and $\alpha$ is a margin parameter. This loss encourages semantic embeddings of neighboring frames to be closer while keeping distant frames apart.

To achieve position invariance, we apply spatial transformations $T$ (e.g., scaling or rotation) to the input frames during training and ensure that the embeddings of the original and transformed frames remain consistent. The spatial consistency loss is defined as:

$$\mathcal{L}_s = \|Z - Z'\|_2, \tag{3}$$

where $Z$ and $Z'$ are the embeddings of the original and transformed frames, respectively.

While the Semantic Alignment Encoder discards position-transformation information, we introduce a separate Transformation Encoder $E_S$ to capture the positional variations (such as translation or scaling) as $S$. The Semantic Alignment Decoder $D$ reconstructs the aligned pose heatmap from the semantic encoding $Z$, and an affine transformation $A$, derived from the transformation encoding $S$, is applied to produce the final pose heatmap $H$.

$$H = A(D(Z), S). \tag{4}$$

### 3.3 TRAJECTORY PROBABILITY DIFFERENCE INTEGRATION

In addition to enforcing semantic continuity, we propose the Trajectory Probability Difference Integration method to ensure temporal continuity in the keypoint distributions. This approach supervises the smooth transition of keypoints across frames by analyzing pixel-level changes in keypoint locations.

For two consecutive frames $I_t$ and $I_{t+\Delta t}$, we define the pixel difference around two keypoint positions, $(x_t, y_t)$ and $(x_{t+\Delta t}, y_{t+\Delta t})$, as:

$$PD((x_t, y_t), (x_{t+\Delta t}, y_{t+\Delta t})) = \sum_{i=-2}^{2} \sum_{j=-2}^{2} |I_1(x_t + i, y_t + j) - I_2(x_{t+\Delta t} + i, y_{t+\Delta t} + j)|. \tag{5}$$

This equation calculates pixel differences in a 5x5 neighborhood around the keypoints to capture local variations. Given the relative stability of lighting and appearance over short periods, the pixel difference $PD$ should remain small for the same keypoint between consecutive frames.

To enforce this, we compute the expected difference in keypoint positions by integrating the keypoint probability heatmaps $H_t$ and $H_{t+\Delta t}$:

$$E = \iint PD((x_t, y_t), (x_{t+\Delta t}, y_{t+\Delta t})) \cdot H_t(x_t, y_t) \cdot H_{t+\Delta t}(x_{t+\Delta t}, y_{t+\Delta t}) \, dx_t \, dy_t \, dx_{t+\Delta t} \, dy_{t+\Delta t}. \tag{6}$$

We minimize the cumulative difference expectation over the trajectory formed by keypoints across frames:

$$\mathcal{L}_t = \int_{t_a}^{t_b} \left| \frac{E(t, t + \Delta t)}{dt} \right| dt. \tag{7}$$

By minimizing $\mathcal{L}_t$, we ensure that keypoints transition smoothly across video frames, promoting temporal coherence in keypoint predictions.

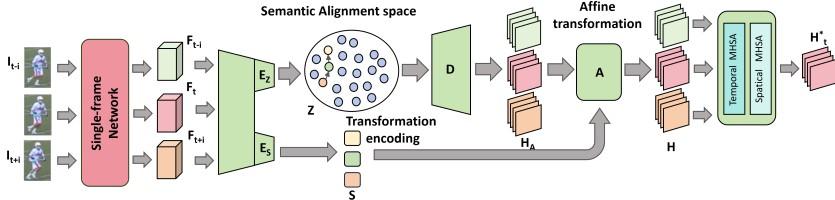

Figure 2: Network architecture of continuity-driven pose estimation model.

### 3.4 NETWORK ARCHITECTURE

In this subsection, we introduce the architecture of our video pose estimation network.

As depicted in Figure 2, the network begins by applying a single-frame backbone to extract features from each individual frame in the video sequence. These features are then encoded into the proposed Semantic Alignment Space. To effectively capture temporal information, a Heatmap Fusion module is employed, which fuses the temporal context from multiple frames to produce the final pose estimation.

Given the success of transformer-based architectures in various computer vision tasks, we adopt the Vision Transformer (ViT) Dosovitskiy et al. (2020) as the backbone of our network. When processing an input image of a human $X = I_t$, it is initially transformed into tokens via a Patch Embedding layer(PE). These tokens are subsequently passed through several transformer layers, each composed of a multi-head self-attention (MHSA) mechanism followed by a feed-forward network (FFN):

$$F_0 = PE(X), F_{n+1}^{'} = F_n + MHSA(LN(F_n)). \tag{8}$$

We input a sequence of video frames $I_{t-i}$ to $I_{t+i}$ into the single-frame pose estimation network, obtaining the corresponding feature maps $F_{t-i}$ to $F_{t+i}$. These feature maps are then passed into the semantic alignment encoder $E_Z$, resulting in the semantic alignment encoding sequence $Z_{t-i}$ to $Z_{t+i}$, and the transformation position encodings $S_{t-i}$ to $S_{t+i}$. Finally, these two components are passed through the heatmap decoder to generate the keypoint probability heatmaps $H_{t-i}$ to $H_{t+i}$ for each frame in the sequence.

To refine the pose estimation at the keyframe $t$, we introduce a Heatmap Fusion module, depicted in Figure 2. This module alternates between two types of self-attention layers: a temporal self-attention layer that fuses information across multiple frames, and a spatial self-attention layer that captures spatial dependencies within each frame. The resulting fused representation is passed through a convolutional layer (the heatmap head), which produces the final keyframe heatmap $H_t^*$.

### 3.5 TRAINING OF VIDEO POSE ESTIMATION MODEL

For training, we utilize contrastive learning losses $\mathcal{L}_s$ and $\mathcal{L}_c$ to ensure a robust learning of the Semantic Alignment Space. To supervise the predicted pose heatmap sequence $H$, we apply a mean squared error (MSE) loss between the predicted heatmaps and the ground truth.

$$\mathcal{L}_{h1} = \sum_{i=t-T}^{t+T} \|G_i - H_i\|_2^2, \tag{9}$$

where $t$ is the keyframe, $G$ represents the ground truth heatmaps, and $T$ defines the temporal window.

Additionally, we introduce a temporal loss $\mathcal{L}_t$ to supervise the pose heatmap sequence, encouraging the continuity of keypoint probability distributions by the pixel continuity of frames. For the final keyframe heatmap $H^*$, we apply an MSE loss after the Heatmap Fusion module to further refine the output:

$$\mathcal{L}_{h2} = \|G - H_t^*\|_2^2. \tag{10}$$

The overall loss function is formulated as follows:

$$\mathcal{L} = \lambda_s \mathcal{L}_s + \lambda_c \mathcal{L}_c + \lambda_{h1} \mathcal{L}_{h1} + \lambda_{h2} \mathcal{L}_{h2}, \tag{11}$$

where $\lambda_s$, $\lambda_c$, $\lambda_{h1}$, and $\lambda_{h2}$ are weights balancing the different loss components.

Table 1: Quantitative results on the PoseTrack17 validation set.

| Method | Backbone | Head | Shoulder | Elbow | Wrist | Hip | Knee | Ankle | **Mean** |
|---|---|---|---|---|---|---|---|---|---|
| PoseTracker Girdhar et al. (2018) | 3D ResNet | 67.5 | 70.2 | 62.0 | 51.7 | 60.7 | 58.7 | 49.8 | 60.6 |
| PoseFlow Xiu et al. (2018) | ResNet-152 | 66.7 | 73.3 | 68.3 | 61.1 | 67.5 | 67.0 | 61.3 | 66.5 |
| FastPose Zhang et al. (2019) | ResNet-101 | 80.0 | 80.3 | 69.5 | 59.1 | 71.4 | 67.5 | 59.4 | 70.3 |
| Simple (R-50) Xiao et al. (2018) | ResNet-50 | 79.1 | 80.5 | 75.5 | 66.0 | 70.8 | 70.0 | 61.7 | 72.4 |
| Simple (R-152) Xiao et al. (2018) | ResNet-152 | 81.7 | 83.4 | 80.0 | 72.4 | 75.3 | 74.8 | 67.1 | 76.7 |
| STEmbedding Jin et al. (2019) | Hourglass | 83.8 | 81.6 | 77.1 | 70.0 | 77.4 | 74.5 | 70.8 | 77.0 |
| HRNet Sun et al. (2019) | HRNet | 82.1 | 83.6 | 80.4 | 73.3 | 75.5 | 75.3 | 68.5 | 77.3 |
| MDPN Guo et al. (2018) | ResNet-152 | 85.2 | 88.5 | 83.9 | 77.5 | 79.0 | 77.0 | 71.4 | 80.7 |
| CorrTrack Rafi et al. (2020) | GoogleNet | 86.1 | 87.0 | 83.4 | 76.4 | 77.3 | 79.2 | 73.3 | 80.8 |
| Dynamic-GNN Yang et al. (2021) | HRNet | 88.4 | 88.4 | 82.0 | 74.5 | 79.1 | 78.3 | 73.1 | 81.1 |
| PoseWarper Bertasius et al. (2019) | HRNet-W48 | 81.4 | 88.3 | 83.9 | 78.0 | 82.4 | 80.5 | 73.6 | 81.2 |
| DCPose Liu et al. (2021) | HRNet-W48 | 88.0 | 88.7 | 84.1 | 78.4 | 83.0 | 81.4 | 74.2 | 82.8 |
| DetTrack Wang et al. (2020) | 3D HRNet | 89.4 | 89.7 | 85.5 | 79.5 | 82.4 | 80.8 | 76.4 | 83.8 |
| FAMI-Pose Liu et al. (2022) | HRNet-W48 | 89.6 | 90.1 | 86.3 | 80.0 | 84.6 | 83.4 | 77.0 | 84.8 |
| TDMI-ST Feng et al. (2023a) | HRNet-W48 | 90.6 | 91.0 | 87.2 | 81.5 | 85.2 | 84.5 | 78.7 | 85.9 |
| DiffPose Feng et al. (2023b) | VIT | 89.0 | 91.2 | 87.4 | 83.5 | **85.5** | **87.3** | 80.2 | 86.4 |
| DSTA He & Yang (2024) | VIT-H | **89.3** | 90.6 | 87.3 | 82.6 | 84.5 | 85.1 | 77.8 | 85.6 |
| ours | HRNet-W48 | 87.1 | 90.3 | 87.5 | 83.7 | 84.4 | 86.7 | 84.2 | 86.1 |
| ours | VIT-B | 87.8 | **91.3** | **88.1** | **84.5** | 84.8 | 87.2 | **85.1** | **87.2** |

Table 2: Quantitative results on the PoseTrack18 validation set.

| Method | Head | Shoulder | Elbow | Wrist | Hip | Knee | Ankle | **Mean** |
|---|---|---|---|---|---|---|---|---|
| STAF Raaj et al. (2019) | - | - | - | 64.7 | - | - | 62.0 | 70.4 |
| AlphaPose Fang et al. (2017) | 63.9 | 78.7 | 77.4 | 71.0 | 73.7 | 73.0 | 69.7 | 71.9 |
| TML++ Hwang et al. (2019) | - | - | - | - | - | - | - | 74.6 |
| MDPN Guo et al. (2018) | 75.4 | 81.2 | 79.0 | 74.1 | 72.4 | 73.0 | 69.9 | 75.0 |
| PGPT Bao et al. (2020) | - | - | - | 72.3 | - | - | 72.2 | 76.8 |
| Dynamic-GNN Yang et al. (2021) | 80.6 | 84.5 | 80.6 | 74.4 | 75.0 | 76.7 | 71.8 | 77.9 |
| PoseWarper Bertasius et al. (2019) | 79.9 | 86.3 | 82.4 | 77.5 | 79.8 | 78.8 | 73.2 | 79.7 |
| PT-CPN++ Yu et al. (2018) | 82.4 | 88.8 | 86.2 | 79.4 | 72.0 | 80.6 | 76.2 | 80.9 |
| DCPose Liu et al. (2021) | 84.0 | 86.6 | 82.7 | 78.0 | 80.4 | 79.3 | 73.8 | 80.9 |
| DetTrack Wang et al. (2020) | 84.9 | 87.4 | 84.8 | 79.2 | 77.6 | 79.7 | 75.3 | 81.5 |
| FAMI-Pose Liu et al. (2022) | 85.5 | 87.7 | 84.2 | 79.2 | 81.4 | 81.1 | 74.9 | 82.2 |
| DiffPose Feng et al. (2023b) | 85.0 | 87.7 | 84.3 | 81.5 | 81.4 | 82.9 | 77.6 | 83.0 |
| TDMI-ST Feng et al. (2023a) | 86.7 | 88.9 | **85.4** | **80.6** | **82.4** | 82.1 | 77.6 | 83.6 |
| ours | **88.1** | **89.5** | 84.9 | 79.9 | 79.8 | **82.9** | 80.9 | **84.1** |

Table 3: Quantitative results on the PoseTrack21 validation set.

| Method | Head | Shoulder | Elbow | Wrist | Hip | Knee | Ankle | **Mean** |
|---|---|---|---|---|---|---|---|---|
| Tracktor++ w. poses Bergmann et al. (2019); Doering et al. (2022a) | - | - | - | - | - | - | - | 71.4 |
| CorrTrack Rafi et al. (2020); Doering et al. (2022a) | - | - | - | - | - | - | - | 72.3 |
| CorrTrack w. ReID Rafi et al. (2020); Doering et al. (2022a) | - | - | - | - | - | - | - | 72.7 |
| Tracktor++ w. corr. Bergmann et al. (2019); Doering et al. (2022a) | - | - | - | - | - | - | - | 73.6 |
| DCPose Liu et al. (2021) | 83.2 | 84.7 | 82.3 | 78.1 | 80.3 | 79.2 | 73.5 | 80.5 |
| FAMI-Pose Liu et al. (2022) | 83.3 | 85.4 | 82.9 | 78.6 | 81.3 | 80.5 | 75.3 | 81.2 |
| DiffPoseFeng et al. (2023b) | 84.7 | 85.6 | 83.6 | 80.8 | 81.4 | 83.5 | 80.0 | 82.9 |
| TDMI-STFeng et al. (2023a) | 86.8 | **87.4** | **85.1** | 81.4 | 83.8 | 82.7 | 78.0 | 83.8 |
| ours | **87.4** | 87.3 | 85.1 | **81.8** | **84.0** | 83.4 | **82.0** | **84.7** |

## 4 EXPERIMENTS

### 4.1 EXPERIMENTAL SETTINGS

#### 4.1.1 DATASETS

The *PoseTrack* benchmark has played a pivotal role in advancing video-based human pose estimation. **PoseTrack17** Andriluka et al. (2018) comprises 250 training video sequences and 50 validation sequences, yielding a total of 80, 144 pose annotations following the standard protocol. This dataset includes 15 keypoints for each annotation, complemented by a joint visibility flag. The subsequent release, **PoseTrack18** Andriluka et al. (2018), significantly expands the dataset, featuring 593 training and 170 validation sequences with 153, 615 pose annotations. The latest iteration, **PoseTrack21** Doering et al. (2022b), builds on the prior version by enhancing the pose annotations, particularly for smaller individuals and those in crowded scenes, resulting in 177, 164 total pose annotations. Notably, PoseTrack21 refines the joint visibility flag, incorporating more detailed occlusion information to improve pose estimation accuracy.

#### 4.1.2 EVALUATION METRIC

We utilize mean Average Precision (mAP) as the primary evaluation metric for pose estimation. The AP is computed for each keypoint, followed by averaging over all keypoints to derive the final mAP score.

#### 4.1.3 IMPLEMENTATION DETAILS

In alignment with previous top-down pose estimation methods Xiao et al. (2018); Wei et al. (2016), each individual is first cropped based on their bounding box during preprocessing. Consistent with common practice Liu et al. (2021; 2022), the cropping area is expanded by 25% beyond the bounding box to include contextual information.

For data augmentation, we apply Random Flip, Half Body Transform, and Random Scale Rotation during training. The AdamW optimizer is used, initialized with a learning rate of $5 \times 10^{-4}$.

During training, we employ the ViT-B architecture as the backbone for single-frame feature extraction, using an input resolution of $256 \times 192$. Optimization is conducted with AdamW, starting with a learning rate of $1 \times 10^{-3}$. The temporal span $T$ for the input frame sequence is set to 2. We initialize the backbone with pre-trained weights from the MS-COCO dataset and train the network for 100 epochs.

For comparison with non-transformer-based approaches, we train an additional version of our model using the HRNetW48 backbone, a well-established architecture for pose estimation. Like the ViT-B backbone, HRNetW48 is initialized with MS-COCO pre-trained weights, and we ensure consistent training settings to enable a fair comparison between the two backbones.

### 4.2 COMPARISON WITH STATE-OF-THE-ART APPROACHES

**Evaluation on PoseTrack2017 Dataset:** On the PoseTrack2017 dataset, our method is assessed against a gamut of other methods, with performance metrics delineated in Table 1. Our model achieves an mAP of 87.2. When compared against the previous TDMI-ST model Feng et al. (2023a), our approach showcases a 0.8 mAP increment. We also compare our method with the latest DiffPose Feng et al. (2023b) and DSTA He & Yang (2024) which also use the Vision Transformer as the backbone network, and the experimental results proved our advantage.

**Evaluation on PoseTrack2018 Dataset:** On progressing to the PoseTrack2018 dataset, the results, collated in Table 2, underscore our model's supremacy. Setting new state-of-the-art results, our model procures an overall mAP of 84.1, surpassing TDMI-ST Feng et al. (2023a) by 0.5 mAP.

**Evaluation on the PoseTrack21 Dataset:** We also conduct a comprehensive evaluation on the PoseTrack21 dataset, with results compiled in Table 3. Baseline performance metrics from existing works Bergmann et al. (2019); Rafi et al. (2020); Doering et al. (2022a) are referenced from the official dataset Doering et al. (2022a). Additionally, we replicate several prominent methods,

Table 4: Complexity comparison with HRNet-W48 backbone.

| Method | Params | reuslt(mAP) |
|---|---|---|
| PoseWarper Bertasius et al. (2019) | 71.1M | 81.0 |
| DCPose Liu et al. (2021) | 65.2M | 82.8 |
| DSTA He & Yang (2024) | 63.9M | 84.6 |
| ours | 64.3M | 86.1 |

Table 5: Ablation study of different combinations of our network.

| Baseline | Semantic Space | Trajectory Probability Difference | Fusion | reuslt(mAP) |
|---|---|---|---|---|
| ✓ | | | | 85.5 |
| ✓ | ✓ | | | 85.8 |
| ✓ | ✓ | ✓ | | 86.7 |
| ✓ | ✓ | ✓ | ✓ | 87.2 |
| ✓ | ✓ | | ✓ | 86.3 |

including DCPose Liu et al. (2021), FAMI-Pose Liu et al. (2022), DiffPose Feng et al. (2023b), and TDMI-ST Feng et al. (2023a), and reevaluate them on this dataset for a more thorough comparison. Our method achieves an mAP of 84.7, outperforming FAMI-Pose Liu et al. (2022) (81.2 mAP) and TDMI-ST Feng et al. (2023a) (83.8 mAP), reinforcing its robustness and establishing its leading performance in this challenging benchmark.

**Complexity comparison with HRNet-W48 backbone:** We conduct experiments to evaluate the computational complexity on the PoseTrack2017 validation set, with the results shown in Table 4. To ensure a fair comparison, we use the same HRNet-W48 backbone. As indicated in Table 4, our method outperforms the latest DSTA approach, achieving superior performance with a similar number of parameters.

### 4.3 ABLATION STUDY

In this section, we first undertake ablation studies to evaluate the contributions of each module within our proposed framework. Additionally, we investigate the different components of the semantic alignment space. Furthermore, we evaluate the impact of different frame intervals in video pose estimation training.

#### 4.3.1 ABLATION STUDY OF DIFFERENT COMPONENTS OF NETWORK

In this section, we validate the effectiveness of different network components by assessing their impact on the overall performance. First, we establish our Baseline method, which consists solely of the single-frame pose estimation network.

Next, we incorporate the Semantic Alignment Space for continuity perception, training the pose estimation network with this added component. We also introduce the Trajectory Probability Difference method to supervise the temporal coherence of the keypoint probability distributions in the video pose estimation network.

Subsequently, we apply the multi-frame heatmap fusion module, which merges heatmaps from multiple frames to generate the final heatmaps. Additionally, we evaluate the network's performance when trained without the Trajectory Probability Difference method to assess its contribution to the model's overall effectiveness.

From the experimental results presented in Table 5, it is evident that the introduction of the pose alignment latent space results in a performance improvement over the Baseline.

Furthermore, the Trajectory Probability Difference method leads to an even more significant enhancement, achieving a 0.9 mAP increase.

Additionally, the integration of the multi-frame heatmap fusion module further boosts the overall performance of the network, with a 0.5 mAP improvement compared to the Baseline.

Table 6: Ablation study of different components of the semantic alignment space.

| $\mathcal{L}_c$ | $\mathcal{L}_s$ | reuslt(mAP) | $Z$ | $S$ | reuslt(mAP) |
|---|---|---|---|---|---|
|  |  | 86.3 | ✓ |  | 12.7 |
| ✓ |  | 86.9 | ✓ | ✓ | 87.2 |
|  | ✓ | 86.4 |  |  |  |
| ✓ | ✓ | 87.2 |  |  |  |

These experimental results validate the effectiveness of each component in our video pose estimation network.

### 4.3.2 ABLATION STUDY OF DIFFERENT COMPONENTS OF THE SEMANTIC ALIGNMENT SPACE

In this section, we conduct a series of ablation experiments to analyze the contribution of various components within the Semantic Alignment Space, as shown in Table 6. Specifically, we evaluate the importance of the semantic continuity loss $\mathcal{L}_c$, which enforces temporal consistency, and the spatial consistency loss $\mathcal{L}_s$, which promotes position invariance. Additionally, we assess the impact of using both the semantic encoding $Z$ and the transformation encoding $S$ on overall performance.

The results show that both $\mathcal{L}_c$ and $\mathcal{L}_s$ positively contribute to the network's performance. However, when applied individually, their impact is limited, or they even negatively affect the network's stability and accuracy.

Moreover, the combination of semantic encoding $Z$ and transformation encoding $S$ is crucial. The network fails to train effectively without the transformation encoding $S$, emphasizing the essential role of encoding positional transformations for proper model convergence and accurate pose estimation.

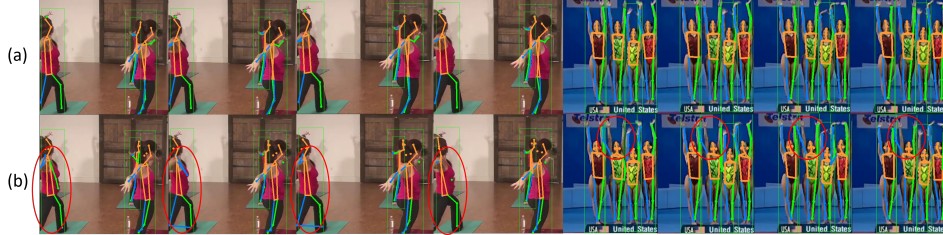

Figure 3: visualization of video pose estimation. (a) shows the predictions of our model, (b) shows the predictions of DCPose.

### 4.4 VISUALIZATION OF VIDEO POSE ESTIAMTION

In this section, we present the visualization of our method's prediction results on the PoseTrack2017 dataset, comparing them with those of the DCpose Liu et al. (2021) method. As illustrated in Figure 3, our model consistently achieves smooth and accurate predictions in sequential scenes. This performance can be attributed to the advantage of incorporating both semantic continuity and distribution continuity supervision in our approach, which ensures temporally coherent pose estimation across frames.

## 5 CONCLUSION

In this paper, we have presented a continuity-driven approach for video-based human pose estimation that improves temporal coherence in keypoint detection across frames. Unlike previous methods, our approach supervises the entire pipeline to ensure both semantic and pixel-wise keypoint continuity. We proposed the Semantic Alignment Space for aligning semantic information across frames and the Trajectory Probability Difference Integration method to ensure smoother keypoint transitions. Our Multi-frame Heatmap Fusion further refines predictions by leveraging information from adjacent frames. Experiments on PoseTrack datasets show that our method consistently outperforms state-of-the-art techniques, enhancing pose estimation accuracy and robustness.

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
