# OpenReview forum: "Continuity-Driven Pose Estimation for Videos"
_ICLR.cc/2025/Conference — ICLR 2025 Conference Withdrawn Submission_

### Official Review · Reviewer_PXFH · 2024-10-28

**Soundness:** 3
**Presentation:** 3
**Contribution:** 3
**Rating:** 5
**Confidence:** 4

**Summary:**

This paper identifies two critical types of continuity for video-based 2D human pose estimation. To this end, they propose Semantic Alignment Space, Trajectory Probability Difference Integration, and Multi-frame Heatmap Fusion module, which capture temporal dependencies and deliver coherent pose estimations across video frames. Comprehensive experiments on benchmark datasets, including PoseTrack17, PoseTrack18, and PoseTrack21, demonstrate the effectiveness of the proposed method.

**Strengths:**

1.The insight of semantic continuity is interesting and is well-illustrated in Figure 1c.

2.The Trajectory Probability Difference Integration showcases some technical.

**Weaknesses:**

1.The statement that "Consequently, the feature fusion process is disconnected from the backbone, which hampers the model’s ability to fully capture and utilize the temporal dynamics of human motion." is not rigorous.

For example, TDMI-ST[1] introduces a Multi-Stage Temporal Difference Encoder to effectively integrate features extracted from different frames within the backbone network. Although the backbone is frozen and not trained, the features used are from different resolutions within the backbone (HRNet). Therefore, I believe this statement is not rigorous.

2.Although the idea of semantic continuity is new and interesting, the concept of pixel-wise keypoint distribution continuity is less intuitive.

The authors say that "...., because it is noticed that the pixel differences for the same keypoint between consecutive frames can be considered approximately invariant, given that lighting conditions are typically stable over short periods.". When a keypoint encounters continuous occlusion, its probability distribution may have significant errors. For example, if we have three pose frames where the first two are normal and the last one is occluded, using TPDI is reasonable. However, if the first two frames are occluded and only the last one is normal, minimizing the expected difference might cause the probability distribution of the last frame, which is originally correct, to move closer to the previous two frames with larger errors.

3.The effectiveness of the backbone.

Unlike previous methods which typically freeze the backbone, the proposed method updates the backbone during the training process. Although the authors claim that "This allows the backbone to learn temporal information alongside the feature fusion network.", there is a lack of intuitive experimental results to support this claim.

We are uncertain whether the improvement gained by training the backbone is due to its learning of temporal correlations or simply to its better fit to the current dataset. As the authors mention in 4.1.3 Implementation Details, the backbones (ViT-B/HRNet) were pre-trained on COCO, while this work uses the PoseTrack dataset.

The ablation study reports a baseline performance of 85.5 using the backbone alone. It's unclear whether this result was obtained with a frozen or a trained backbone.

To further elucidate the impact of training the backbone, I suggest that the authors conduct both qualitative and quantitative comparisons. For the qualitative comparisons, visualizing the feature of the intermediate layers of the backbone could provide valuable insights into the differences between the frozen and trainable models. For the quantitative comparisons, I suggest freezing your updated backbone and replacing the backbone in the previous method to see if it can improve the performance of the previous method.

[1] Mutual Information-Based Temporal Difference Learning for Human Pose Estimation in Video CVPR'23

4. The comparisons in Tables 1, 2, and 3 include only one study published in 2024, which does not sufficiently demonstrate the effectiveness of the proposed method.

**Questions:**

Please see the Weaknesses.

While it is challenging to attribute the performance improvement solely to the backbone's ability to learn temporal correlations or its adaptation to the specific dataset (as the enhancement is likely a result of both factors), a more in-depth analysis of this issue could contribute to a more comprehensive understanding of the proposed method and improve the quality of this paper.

---

### Official Review · Reviewer_wJHG · 2024-11-02

**Soundness:** 2
**Presentation:** 2
**Contribution:** 3
**Rating:** 5
**Confidence:** 5

**Summary:**

This paper presents a method called Continuity-Driven Pose Estimation, which aims to improve the accuracy and stability of pose estimation by reinforcing continuity between poses. The method applies a novel continuity constraint strategy to help the model better predict human poses in complex motion scenes. The authors validate the method's effectiveness through experiments and demonstrate superior performance compared to existing methods.

**Strengths:**

**Originality**: The paper introduces the concept of enhancing pose estimation with continuity-driven strategies, which is conceptually novel. Using continuity constraints to smooth predictions across frames could have practical value in complex dynamic scenes.

**Experimental Performance**: The method was evaluated on multiple datasets, showing some advantages in specific metrics compared to baselines.

**Weaknesses:**

**Unclear Problem Definition**: The paper does not clearly define whether it addresses 2D human pose estimation or single-view 3D human pose estimation. These are distinct tasks, especially for 3D pose estimation, where single-view and multi-view pose estimation present different challenges and require separate approaches.

**Evaluation Metrics**: The evaluation metrics used in the paper are primarily suited for 2D pose estimation, whereas the experimental datasets and comparison methods focus on 3D pose estimation tasks. For 3D pose estimation, commonly used metrics include MPJPE (Mean Per Joint Position Error), which is notably absent.

**Limited Dataset Representation**: The selection of datasets for the experiments is not sufficiently representative. For 2D pose estimation, results on widely used datasets like COCO and MPII should be included. For 3D pose estimation, results on benchmark datasets such as Human3.6M, CMU-Panoptic, and MPI-INF-3DHP are needed. As the paper compares methods that have conducted experiments on these datasets, adding results on these benchmarks would improve the study's relevance and comparability.

**Lack of Detail in Methodology and Experiments**:

- The discussion of related work is not in-depth, and the paper does not highlight the limitations of existing methods.
- The method section lacks sufficient theoretical explanation and details on network architecture.
- The experimental comparisons could be enhanced by including methods like GLA-GCN (Global-local Adaptive Graph Convolutional Network for 3D Human Pose Estimation from Monocular Video).
Some existing studies on joint smoothing methods are available in the literature. Comparative experiments with some of these established models would be beneficial.
- Adding targeted analysis and specific improvements could lead to better overall results for joints where performance decreases.

**Questions:**

**Problem Definition**: Could the authors clarify whether the paper focuses on 2D or 3D pose estimation? If it is 3D, is it single-view or multi-view, as these tasks require different approaches?

**Evaluation Metrics**: Why were the chosen evaluation metrics primarily suited for 2D pose estimation instead of using MPJPE, a standard metric for 3D pose estimation? Would re-evaluating using MPJPE provide more insights into model performance?

**Dataset Selection**: Could the authors consider adding results from more representative datasets? COCO and MPII are widely recognized for 2D pose estimation, while Human3.6M, CMU-Panoptic, and MPI-INF-3DHP are commonly used for 3D pose estimation, which could enhance the study’s relevance.

**Methodology and Experiment Details**:

- Could the authors expand the related work discussion to outline the limitations of current approaches better?
- The methodology section lacks detail. Could additional architectural information be provided to enhance reproducibility?
- Would the authors consider including GLA-GCN or other recent methods for comparison to better contextualize the performance?
- Have the authors considered incorporating existing models for comparison for joint smoothing methods?
- Could the authors provide targeted analyses or improvement strategies to boost these results for joints with decreased performance?

---

### Official Review · Reviewer_G2SM · 2024-11-02

**Soundness:** 3
**Presentation:** 2
**Contribution:** 2
**Rating:** 5
**Confidence:** 3

**Summary:**

This paper proposes two mechanisms to improve the consistency of the learned representation for each video frame: semantic alignment and keypoint continuity. In addition to these, a multi-frame heatmap fusion block is designed to refine the learned heatmap and improve temporal consistency. The proposed method achieves state-of-the-art performance on three datasets, and the effectiveness of each component is verified by an ablation study.

**Strengths:**

1. This paper presents comprehensive experiments and ablation studies.
2. The proposed method achieves state-of-the-art performance on three datasets.

**Weaknesses:**

1. Compared to previous approaches, the proposed method finetunes the single-frame feature backbone. It remains unclear if this brings large improvements. Can authors do an ablation study where the single-frame feature backbone is frozen? In addition, it'd be interesting to compare with another baseline method, fine-tuning its feature encoder as well (if it was not fine-tuned originally).
2. From Tab. 5, it looks like the performance gain from semantic alignment is quite marginal.
3. According to Tab. 6, removing $S$ significantly decreases the performance. Can authors explain why that is? Maybe this is related to my first question in the following block.

**Questions:**

1. How does the Transformation Encoding $S$ work? How are features from two branches merged together (into a single feature map $H$)? Can the authors provide more details?

---

### Note · Authors · 2024-11-13

I have read and agree with the venue's withdrawal policy on behalf of myself and my co-authors.